# Structural rearrangements of the histone octamer translocate DNA

Silvija Bilokapic [1], Mike Strauss[2] & Mario Halic [1]

Nucleosomes, the basic unit of chromatin, package and regulate expression of eukaryotic genomes. Nucleosomes are highly dynamic and are remodeled with the help of ATP-dependent remodeling factors. Yet, the mechanism of DNA translocation around the histone octamer is poorly understood. In this study, we present several nucleosome structures showing histone proteins and DNA in different organizational states. We observe that the histone octamer undergoes conformational changes that distort the overall nucleosome structure. As such, rearrangements in the histone core α-helices and DNA induce strain that distorts and moves DNA at SHL 2. Distortion of the nucleosome structure detaches histone α-helices from the DNA, leading to their rearrangement and DNA translocation. Biochemical assays show that cross-linked histone octamers are immobilized on DNA, indicating that structural changes in the octamer move DNA. This intrinsic plasticity of the nucleosome is exploited by chromatin remodelers and might be used by other chromatin machineries.

---

[1] Gene Ceter, Department of Biochemistry, University of Munich LMU, 81377 Munich, Germany. [2] Cryo EM Facility, Max Planck Institute for Biochemistry, 82152 Martinsried, Germany. Correspondence and requests for materials should be addressed to M.H. (email: halic@genzentrum.lmu.de)

The packaging of DNA into chromatin regulates access to genetic material. The basic building block of chromatin is the nucleosome, which contains ~ 150 base pairs (bp) of DNA wrapped around an octamer composed of two copies each of histones H2A, H2B, H3, and H4[1–3]. The nucleosome structure is stabilized by electrostatic interactions between the phosphate backbone of the DNA and positively charged residues on histones[1].

Nucleosomes are highly dynamic and can move along the DNA in an uncatalyzed or chromatin remodeling enzyme-driven way[4,5]. Uncatalyzed nucleosome sliding is an intrinsic property of nucleosomes and resembles the sliding achieved by chromatin remodeling enzymes[4,6]. At increased temperatures or when contacts of the histone core with the DNA are weakened, nucleosomes display high uncatalyzed mobility[7–10]. Nucleosomes are positioned on the DNA with the help of chromatin remodeling enzymes that use the energy from ATP hydrolysis to move them in vivo[4,6,11]. Different subfamilies of chromatin remodelers catalyze various nucleosomal transformations. Some chromatin remodelers can change the composition of the histone octamer, whereas others slide nucleosomes without disassembling the octamer[6]. It has been shown that imitation switch (ISWI) chromatin remodelers can conformationally rearrange the histone octamer and such rearrangement is essential for translocating DNA[6,12]. These results are consistent with prior observations showing flexibility of a nucleosome in the context of transcription and nucleosome assembly[13–19]. Our recent structures of nucleosomes provided a mechanistic explanation of DNA unwrapping that is utilized by RNA polymerase II and other DNA-based enzymes[20].

Nucleosome positioning has a central role in transcription, DNA replication, repair, and recombination. Mutations in chromatin remodelers are strongly associated with, or even drive cancers, highlighting the importance of nucleosome organization in genome stability[21,22]. Despite the importance of nucleosome remodeling, relatively little is known about the structural mechanisms of histone octamer translocation on the DNA. In this work, we solved cryo-electron microscopy (EM) structures with differently organized histone octamers and DNA. Our structures show conformational changes in the histone octamer that are required for DNA translocation and thermally driven nucleosome sliding. We propose that the same intrinsic property of the nucleosome is also utilized by chromatin remodeling enzymes[12].

## Results

**Cryo-EM structure of nucleosome core particles.** We collected cryo-EM data of nucleosome core particles (NCPs) assembled on a 601 DNA at physiological conditions (Supplementary Fig. 1a–c)[23,24]. In the electron micrographs, NCPs are present in various orientations: as a disk, tilted views, and side views (Supplementary Fig. 1d, e). In two-dimensional (2D) class averages, high-resolution details such as the DNA dyad, major and minor DNA grooves, and histone α-helices are visible (Supplementary Fig. 1e). We solved the structure of the NCP to 4.5 Å resolution using all particles (Supplementary Fig. 1f-h). Further classification of NCPs revealed several defined classes. The first class (Class 1—canonical nucleosome) is resolved to an average resolution of 3.8 Å (Fig. 1a, Supplementary Fig. 2a-c and Table 1) and is similar to previous structures of the NCP[1,25–27]. The second class (Class 2—distorted nucleosome) was reconstructed to 4.0 Å and shows a nucleosome with differently organized DNA (Fig. 1a and Supplementary Fig. 2d-f). Although we used a high-affinity binding and positioning 601 DNA sequence, the histone octamer is better resolved than the DNA in both classes (Supplementary Fig. 2g),

suggesting the DNA is at least partially mobile around the NCP. We also observe that ~ 15 % of nucleosome particles have unwrapped DNA in agreement with the structures we have recently described[20].

We compared the Class 1 and Class 2 cryo-EM structures to the previously determined X-ray structure of the NCP assembled on 601 DNA (PDB:3LZ1)[27]. We observed that the X-ray structure fits well into the Class 1, but not into the Class 2 structure

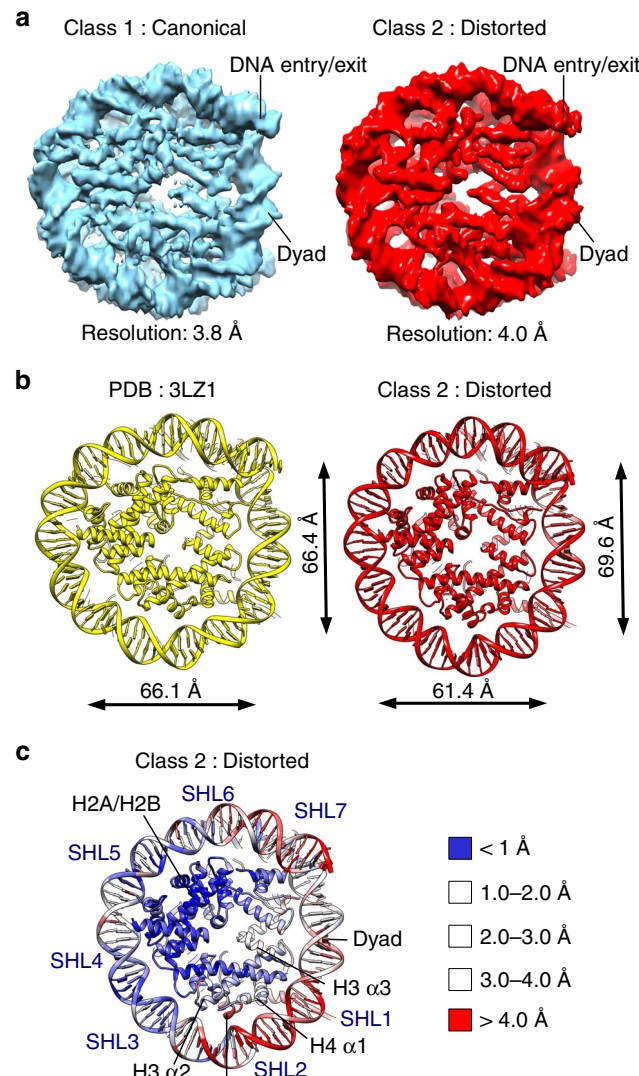

**Fig. 1** Structural plasticity of the nucleosome core particle (NCP). **a** Cryo-EM maps of the NCP in two distinct conformations. Class 1 (left, blue) resembles canonical nucleosome, whereas Class 2 (right, red) is in distorted conformation. Class 1 is resolved to 3.85 Å and Class 2 to 4.05 Å (0.143 cutoff in FSC curve). Class 1 contains 51 000 particles and Class 2 contains 58 000 particles. **b** Global changes in the nucleosome structure. Comparison of X-ray structure (PDB:3LZ1) and Class 2 (distorted nucleosome) models. The nucleosome core particle contracts along the symmetry axis by 8% (distance between nucleotides 2 and 38) and expands in the perpendicular direction by 5% (distance between nucleotides 17 and 58). **c** RMSD between the X-ray structure (PDB:3LZ1) and the Class 2 model, showing the extent of rearrangements in the NCP. The X-ray structure and the Class 2 model were superimposed and RMSD of Cα was calculated and depicted. DNA at SHL 1–2 and SHL 6–7 shows the largest movements between these two structures (> 4 Å). H3 α1, α2, and α3 show the largest rearrangements in the histone octamer

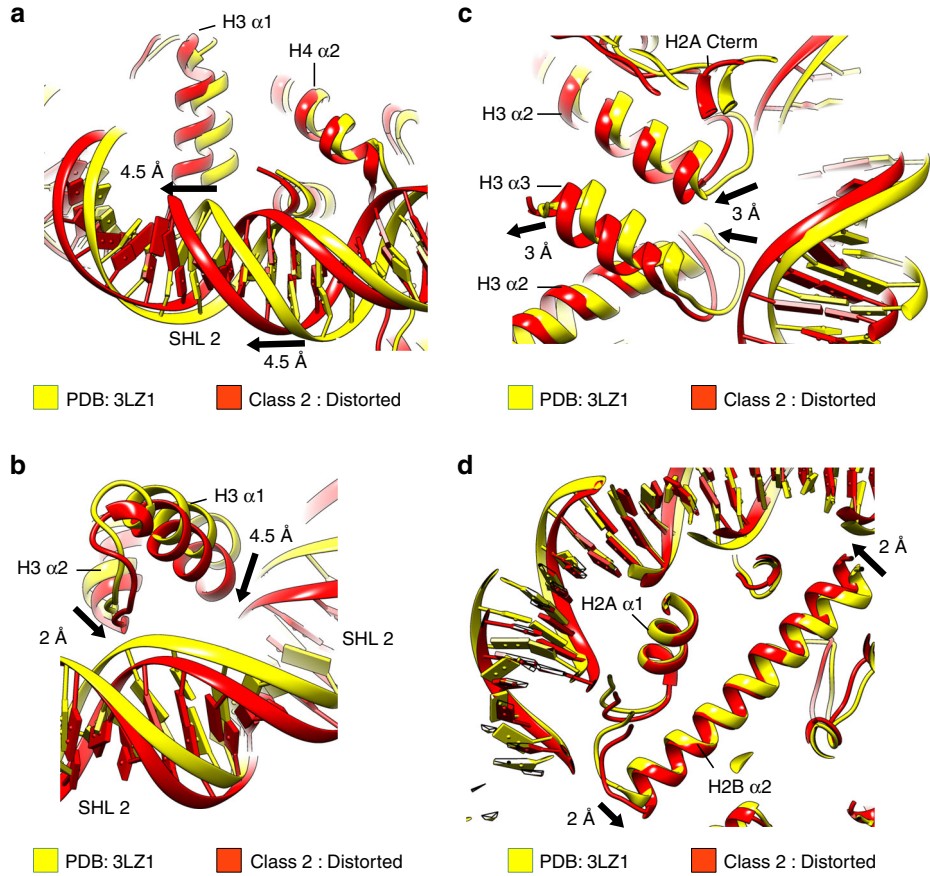

**Fig. 2** Structural rearrangements in the histone core. Comparison of the X-ray structure (PDB:3LZ1, yellow) and the Class 2 model (red). Arrows depict the direction of the helix movements. The degree of movement between X-ray structure and the Class 2 model is shown, rounded to half an Å. **a** Conformational rearrangement of H3 α1 and DNA at SHL 2. H3 α1 moves 4.5 Å in Class 2 when compared with the X-ray structure. DNA at SHL 1.5 moves 4.5 Å towards SHL 2. **b** Conformational rearrangement of the H3 α2, H3 α1, and DNA at SHL 2. The H3 α2 tilts by 2 Å at its N-terminal end. The H3 α1 moves 4.5 Å. H3 α1 and H3 α2 move toward each other and push the DNA at SHL 2.5 outward. **c** Conformational rearrangement of H3 α2, H3 α3, and DNA at the dyad. H3 α2 tilts and moves inward by 3 Å at its C-terminal end. H3 α3 also moves inward by 3 Å. Concomitantly, the DNA at the dyad moves more than 3 Å toward the centre of the nucleosome. **d** Conformational rearrangement of H2B α2. H2B α2 tilts by 2 Å toward the center of the nucleosome at SHL 3.5 and 2 Å away from the center at SHL 5.5. This pushes the DNA at SHL 5.5 outward and pulls the DNA at SHL 3.5 inward

(Supplementary Fig. 2h). A closer examination revealed that most of the DNA and many histone core α-helices are in a different conformation in Class 2 than in the X-ray structures (Supplementary Fig. 2h). For comparison, we also fitted the X-ray structure to the previously published cryo-EM map of the nucleosome (EMD-8140)[26]. Contrary to the Class 2 map, the X-ray structure fits well into this cryo-EM map[26]. The observed differences might be explained by the buffer conditions that were used. To obtain the EMD-8140 cryo-EM map, nucleosomes were prepared without any salt, whereas our nucleosomes were prepared at more physiological conditions. In our recent work[20] we have solved the structure of the nucleosome at low-salt condition to 3.7 Å (Class LS). In this dataset, most wrapped nucleosome particles were in the canonical nucleosome conformations[20], consistent with the previously published cryo-EM map of the nucleosome[26]. Our data suggest that at physiological conditions a fully wrapped nucleosome is more dynamic and can be found in multiple conformations.

**Nucleosome distortion.** We refined the X-ray model of the nucleosome for both Class 1 and Class 2 cryo-EM maps (Fig. 1b and Supplementary Fig. 3a, b). In both classes, the resolution of

the histone octamer is 4.0 Å or better with many side chains resolved (Supplementary Fig. 3a). This allowed us to build an accurate model (Supplementary Fig. 3c). The refined model for Class 1 is highly similar to the X-ray structure (PDB:3LZ1), indicating that this cryo-EM map resembles the canonical nucleosome conformation (Supplementary Fig. 3b). We superimposed X-ray (PDB:3LZ1) and the Class 2 cryo-EM models, and observed many differences. First, the comparison of these models revealed global changes in the nucleosome structure. We observed that the nucleosome shrinks along the symmetry axis and expands in the perpendicular direction (Fig. 1b). Although in the X-ray structure the distance from the dyad to the opposite side of the histone octamer is 66 Å, this distance shrinks in Class 2 to 61 Å. Concomitantly, the perpendicular distance between the superhelices[1,2] SHL 2 and SHL 6 expands from 66 Å in the X-ray structure to 70 Å in Class 2 (Fig. 1b). Our data show that the nucleosome is highly dynamic in solution. The nucleosome contracts 8% along the symmetry axis and expands 5% in the perpendicular direction. It has also been proposed that nucleosomes in solution might exist in multiple states of DNA conformation, with only some of these having been seen in the X-ray structures[28–30].

We calculated the root-mean-square deviation (RMSD) of the X-ray structure and cryo-EM maps to depict regions that display highest movements. Although in Class 1 and Class LS[20] structures we observe only minor movements (Supplementary Fig. 3d), the RMSD data show larger rearrangements in the Class 2 structure (Fig. 1c). We observe that the DNA and histones at SHLs 1.5–2.5 and at the entry/exit site (SHL 6–7) show the highest variability between these two structures (Fig. 1c). Our data show that the nucleosome can adopt multiple conformations providing a structural basis for nucleosome plasticity. The nucleosome plasticity may contribute to chromatin regulation in many processes.

**Structural rearrangement of the histone octamer**. Concomitantly, with the structural changes in the DNA, we observe rearrangements of many α-helices in the histone core. In particular, the histone H3 α1 and α2 helices display strong movements in the Class 2 structure when compared with the X-ray, Class LS[20], and Class 1 structure (Fig. 1c). Our structures show that H3 α1 moves 4.5 Å in the Class 2 map compared with the X-ray structure. H3 α1 coordinates the DNA at SHL 1.5 and the movement of H3 α1 also pushes the DNA more than 4.5 Å toward SHL 2.5 (Fig. 2a). H4 α1 coordinates the DNA at SHL 1.5 together with H3 α1 and moves in the same direction as H3 α1. Although H3 α1, H4 α1, and the DNA at SHL 2 move by > 4 Å, the contacts between histones and DNA stay preserved in the Class 2 structure.

In the Class 1 and Class LS[20] structures, the N-terminal tail of histone H4 interacts with the DNA at SHL 2.5 (Supplementary Fig. 4a). As DNA slides away in the Class 2 structure, the interaction of the H4 N-terminal tail with the DNA at SHL 2.5 is lost. This is most likely, because the positively charged H4K16 and H4R17 are too distant from the DNA phosphate backbone of SHL 2.5, which moves > 4 Å in the Class 2 structure. This leads to the rearrangement of the H4 tail, which is more flexible in the Class 2 structure and inserts into the major grove of the DNA at SHL 2 (Supplementary Fig. 4a). The H4 tail insertion also results in widening of the major groove at SHL 2 by > 1 Å and suggests that the H4 tail might interact with bases as predicted by molecular dynamics simulations[31]. Deletion of the H4 tail was shown to reduce uncatalyzed histone octamer sliding on the DNA[6,32]. This suggests that the H4 tail contributes to the octamer movement on DNA, either by promoting movement or by stabilizing a distinct conformation of the DNA.

Concomitantly, with the movement of H3 α1, we also observed rearrangement of the long H3 α2 helix that spans from SHL 0.5 to SHL 2.5. In the Class 2 structure H3 α2 is tilted (Fig. 2b, c and Supplementary Fig. 4b). The loop connecting the H3 α1 and α2 helices interacts with the DNA at SHL 2.5, whereas the loop connecting helices α2 and α3 interacts with the DNA near the dyad. In the Class 2 structure, H3 α2 tilts more than 2 Å at its N-terminal end, which pushes the DNA at SHL 2.5 outward and in the direction of SHL 2 (Fig. 2b). Simultaneously, the C-terminal end of H3 α2 tilts in the opposite direction, pulling the DNA at SHL 0.5 inward for > 3 Å (Fig. 2c and Supplementary Fig. 4b). This also leads to inward movement of H3 α3 and the DNA at the dyad (Fig. 2c). Our data raise the possibility that H3 α2 serves as a lever that pushes the DNA at one end and pulls the DNA at the opposite end (Fig. 2b, c and Supplementary Fig. 4b).

We also observed that H4 α2 bends outward at SHL 2.5 to compensate for the nucleosome contraction (Supplementary Fig. 4c). The H4 Loop 2 connects H4 α2 and α3, and interacts with the DNA at SHL 2.5, and movement of the H4 α2 pushes the DNA at SHL 2.5 outward by 3 Å (Supplementary Fig. 4c).

Although movements were most pronounced for histone H3 around SHL 2, we also observed smaller rearrangements in H2A/H2B helices (Fig. 1c). At SHL 3.5, the long H2B α2 and H2A α2 helices tilt toward the dyad and pull the DNA inward (Fig. 2d and Supplementary Fig. 4d). At SHL 5.5, the H2B α2 and H2A α2 helices tilt and push the DNA outward (Fig. 2d and Supplementary Fig. 4d). These conformational rearrangements of the histones H2A/H2B promote nucleosome contraction at SHL 3.5 and stretching at SHL 5.5. In Class 2, the H2B C-terminal helix also moved outward. We also observed that the nucleosome conformation in Class 2 favors insertion of the H2A N-terminal tail at SHL 4.5 (Supplementary Fig. 4e). It has been shown that the deletion of the H2A N-terminal tail increases uncatalyzed nucleosome mobility[32], indicating that the H2A tail stabilizes the nucleosome and DNA in the remodeled conformation.

Our data show that structural rearrangements of the histone core α-helices move DNA in the nucleosome and lead to the nucleosome distortion. These movements are coordinated by tilts of the long α2 helices of H3, H2A, and H2B, which interact with the DNA at two distant locations. These helices serve as levers that push the DNA on one side and pull the DNA on the other side. The long H4 α2 helix bends and pushes the DNA outward to compensate for the nucleosome contraction along its length. Our structures show that rearrangement of the histone octamer contracts the nucleosome along the symmetry axis and expands it in the perpendicular direction (Fig. 1b).

**DNA distortion at SHL 2.5**. In the Class 2 structure, we observed the movement of DNA with the largest movements in the SHL 2 region (Fig. 1c). Our structures show that at SHL 1.5 the DNA is moved away from the dyad and in the direction of SHL 2. Simultaneously, DNA at SHL 3.5 is moved in the direction of the

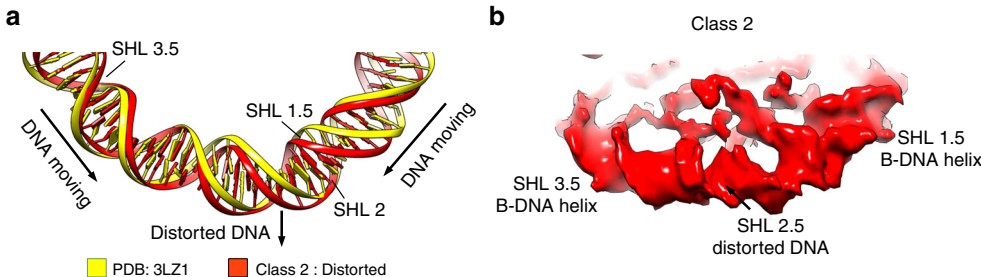

**Fig. 3** Structural rearrangement of the histone core distorts the DNA at SHL 2. **a** Depiction of the DNA organization in the nucleosome in the X-ray structure and Class 2 cryo-EM map at SHL 2. At SHL 1.5, the DNA moves outward and toward SHL 2.5. At SHL 3.5, DNA is pulled inward and toward SHL 2.5. **b** Close-up view at SHL 2.5 in the Class 2 cryo-EM map. In the Class 2 map, the DNA at SHL 2.5 is less defined and does not resemble a B-DNA helix, indicating distortion in the DNA structure at SHL 2.5. At SHL 1.5 and SHL 3.5, the DNA is well resolved and resembles a B-DNA helix

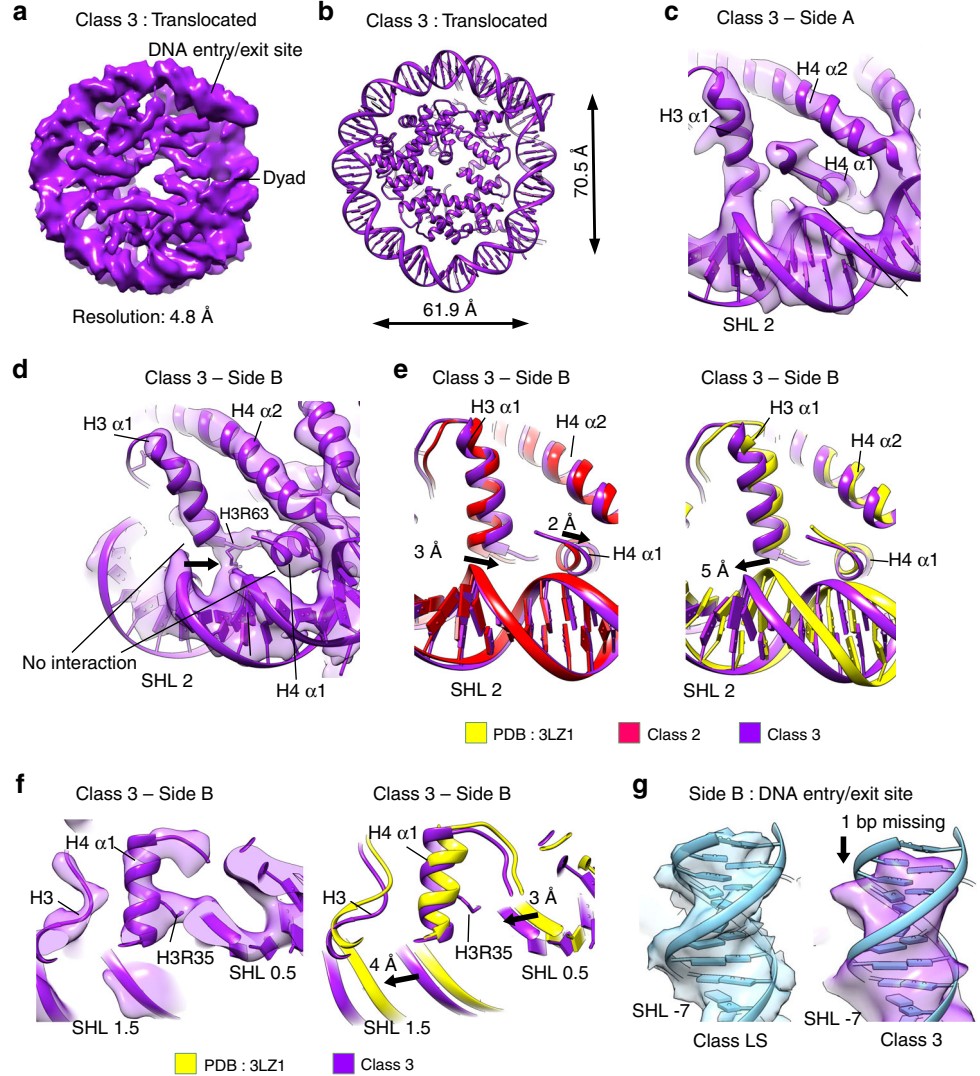

**Fig. 4** Conformational changes in the histone octamer translocate DNA. **a** Cryo-EM map of the NCP Class 3 (translocated nucleosome) at 4.8 Å (0.143 FSC cutoff). Class 3 contains 39 000 particles. **b** Global changes in the Class 3 structure. The histone octamer in the Class 3 NCP is 62 Å wide along the symmetry axis (distance between nucleotides 2 and 38) and 70 Å in the perpendicular direction (distance between nucleotides 17 and 58). **c**, **d** Fitting of the Class 3 (purple) model into the Class 3 map. On the side A, the H3 α1 interacts with the left DNA strand of the superhelix SHL 1.5 **c**. On the side B, the H3 α1 interacts with the right DNA strand of SHL 1.5 **d**. The H4 α1 is detached from the DNA at SHL 1.5 on both sides. **e** Comparison of the X-ray structure (yellow), the Class 2 (red), and the Class 3 models (purple). The degree of movement between the Class 2 and the Class 3 models is shown. The H3 α1 moves back 3 Å in the Class 3 compared with the Class 2. The H3 α1 reverts to similar position that it occupies in X-ray structures, but now it interacts with another DNA strand. DNA at SHL 1.5 moves 5 Å toward SHL 2 compared with X-ray structures. The H4 α1 moves back 2 Å in the Class 3 compared with the Class 2. **f** The H4 α1 detached from the DNA at SHL 1.5 and interacts with the DNA at SHL 0.5. DNA at SHL 1.5 moved 4 Å toward SHL 2 compared with X-ray structures and detaches from the H4 α1. DNA at SHL 0.5 moved 3 Å toward SHL 2 compared with X-ray structures and now interacts with the H4 α1. **g** The model for Class LS_C1 (Class LS reconstructed with C1 symmetry) was fitted into Class LS_C1 and Class 3 cryo-EM maps. In the Class 3 map, DNA at one entry/exit site is shorter than in Class LS_C1 map, indicating that ~1 bp of DNA moved toward the other end

dyad and toward SHL 2 (Fig. 3a). This generates tension in the DNA structure at SHL 2.5. In the Class 2 cryo-EM map, at SHL 1.5 and SHL 3.5, DNA is well resolved with the typical appearance of a B-DNA helix. At SHL 2.5, however, the DNA is distorted and does not resemble a B-DNA helix anymore, suggesting that the DNA might have adopted another conformation at this location (Fig. 3b and Supplementary Fig. 4f). We observed that in the cryo-EM map of Class 2, the density at SHL 2.5 is smaller when compared with other SHLs (Fig. 3b and Supplementary Fig. 4f). This suggests that less DNA might be coordinated at SHL 2.5 in the Class 2 structure, and that several DNA bp might be translocated. Consistently, in X-ray structures of the NCP with

DNA of different length and sequence, SHL 2 and SHL 5 could accommodate a difference of 1 bp[28,29,33,34]. Our data show that movement of DNA at SHL 1.5 and SHL 3.5 toward SHL 2.5 leads to distortion of the DNA. Although our resolution is not sufficient to observe DNA movement at the nucleotide level, our data suggest that the strain created by histone movements at SHL 1.5 and 3.5 might generate sufficient force to move several bp of DNA at SHL 2.

**DNA translocation by the histone octamer.** In the structure of the distorted nucleosome (Class 2), α-helices of the histone octamer rearranged but remained attached to the same DNA

SHL. This class reveals intrinsic nucleosome plasticity and shows how the octamer can be distorted without breaking contacts with the DNA. In addition to the structures of the distorted nucleosome (Class 2), we obtained a nucleosome structure that shows detachment of histone α-helices from the DNA and their translocation on the DNA (Class 3—translocated nucleosome) (Fig. 4).

The Class 3 cryo-EM map is resolved to 4.8 Å, with the resolution of the histone core between 4.5 Å and 5.0 Å, and the DNA 5.0–5.5 Å (Fig. 4a and Supplementary Fig. 5a-e). The overall structure of the nucleosome in the Class 3 cryo-EM map resembles Class 2 structure (Figs. 1b and 4b). The distance from the dyad to the opposite side of the histone octamer is 62 Å,

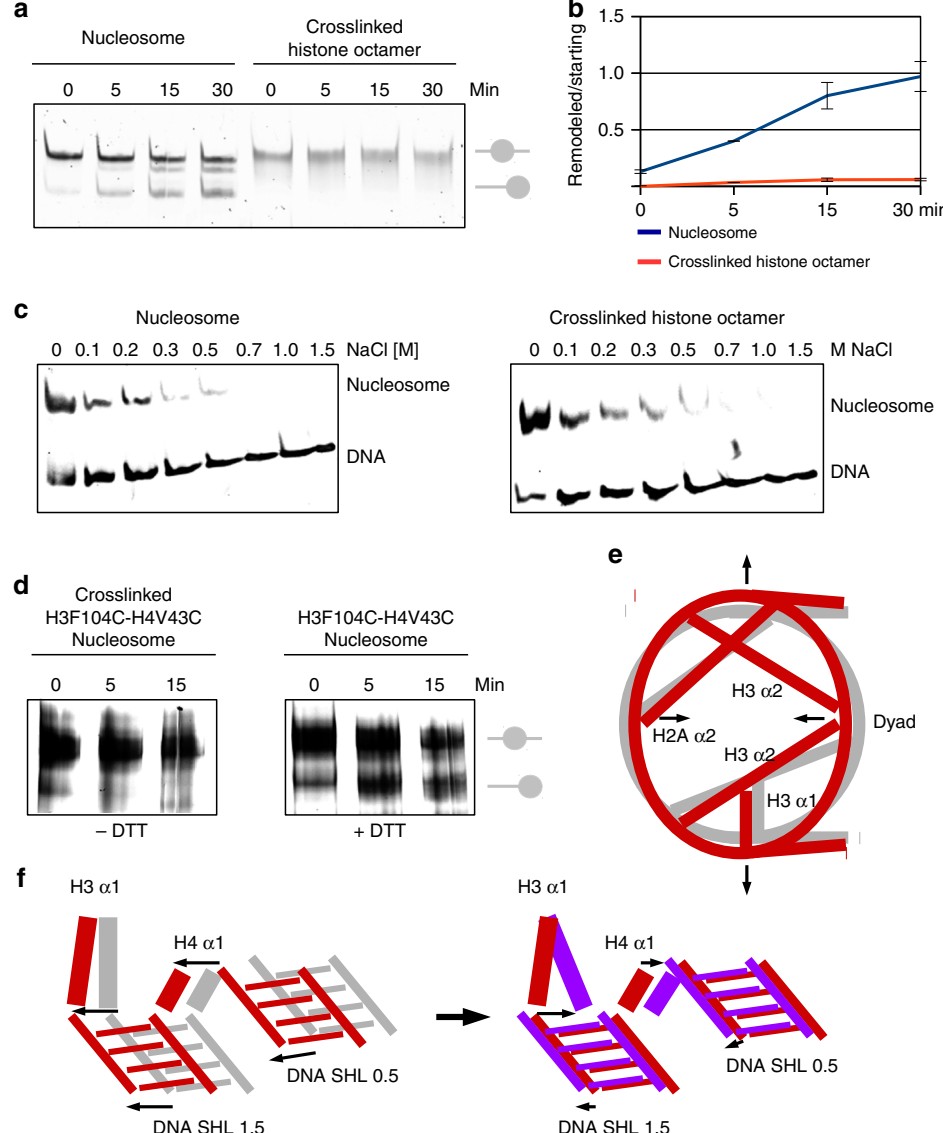

**Fig. 5** Conformational rearrangement of the histone octamer is required for nucleosome sliding. **a** Thermal mobilization of nucleosomes on 227 bp DNA sequence containing 601 sequence in the middle. Native nuclesomes are re-positioned at 60 °C. Nucleosomes with cross-linked octamer did not move. **b** Quantification of three independent thermal shift assays showing that the cross-linked octamer does not move on DNA. The band intensity was quantified at starting and remodeled position at each time point. SD of three independent experiments is shown. **c** Salt-induced disassembly of native and cross-linked nucleosomes. Cross-linked nucleosomes disassemble at elevated salt concentration, indicating that histone octamer did not cross-link with the DNA. **d** Thermal mobilization of nucleosomes on 227 bp DNA sequence containing 601 sequence in the middle. NCP with the disulfide bridge between H3 F104C and H4 V43C is immobilized in the thermal shift assay. Upon addition of the reducing agent (DTT), nucleosomes are mobile again. **e** Model showing the conformational changes in the histone octamer that lead to nucleosome distortion. Tilting of the long α2 helices of H3, H2A, and H2B contracts the nucleosome along the symmetry axis and stretches the nucleosome in the perpendicular direction. This rearrangement of the octamer also moves the DNA. DNA gyres move by > 4 Å at SHL 1.5–2.5 and SHL 5.5–6.5. Canonical nucleosome is shown in gray, distorted in red. **f** Model showing DNA translocation at SHL 2. In the first step of DNA translocation, DNA and the H3/H4 α1 helices move > 4 Å, leading to nucleosome distortion (Class 2). In the next step, the H3 α1 dissociates from the DNA and translocates to DNA strand, which was previously bound by the H4 α1. The H4 α1 detaches from the DNA at SHL 1.5 and binds the DNA at SHL 0.5. This leads to translocation of the DNA by the histone octamer at SHL 2. The DNA is pushed further and moves by > 5 Å when compared with X-ray structures. Arrows indicate direction of the movement. Canonical nucleosome is shown in gray, distorted in red, and translocated in purple

whereas the perpendicular distance between the SHL 1.5 and SHL 5.5 is 70 Å (Fig. 4b). The most pronounced changes between the Class 2 and the Class 3 structures are again at SHL 2. In the Class 3 structure, we observe distinct movements of the histone octamer and the DNA on each side of the nucleosome. On side A of the Class 3 nucleosome, DNA at SHL 1.5 and H3 α1 are in a conformation that is highly similar to the Class 2 structure (Fig. 4c and Supplementary Fig. 5f, g). In this conformation, the H3 α1 helix and the DNA are moved but preserve the contact (Fig. 4c). On side B of the Class 3 structure, the H3 α1 helix moved back for 3 Å and reverted to a similar position that it occupied in the X-ray structures (Fig. 4d, e), whereas the DNA moved forward for 5 Å, indicating a movement of ~ 1 bp. This leads to detachment of the H3 α1 from the DNA and its translocation to another DNA strand of the same SHL. The interaction of H3R69 and H3L65 with the DNA is lost and stronger interaction is now formed by H3R63 (Fig. 4c, d). In Class 3, side B, the H3 α1 interacts with the DNA strand, which interacted with the H4 α1 in previous nucleosome structures. The H4 α1 helix also moved back in the Class 3 structure and detached from the DNA at SHL 1.5 (Fig. 4d-f). The contacts of H4 with SHL 1.5 are lost and primary contact is now formed with the DNA at SHL 0.5. The DNA at SHL 0.5 moved 3 Å toward the H4 α1 and H4R35 makes the main contact of the H4 α1 with the DNA (Fig. 4f and Supplementary Fig. 5h). This translocated H4 α1 from SHL 1.5 to SHL 0.5.

The rearrangement of the H3 α1 leads to further movement of the DNA at SHL 1.5 toward SHL 2.5. In the Class 3 structure, the DNA at SHL 1.5 moved for > 5 Å when compared with the X-ray structures (Fig. 4e). We also observe a further tilt of the H3 α2 at the side B, but not on side A of the Class 3 structure (Supplementary Fig. 5i). The H3 α2 tilts for 3.5 Å at its N-terminal part that interacts at SHL 2.5 when compared with the X-ray structures (Supplementary Fig. 5i). This puts even more strain on the DNA at SHL 2 region and leads to even greater distortion of DNA at SHL 2 and SHL 3. Similar to the Class 2 structure, in the Class 3 structure the H4 tail inserts into the major grove of SHL 2 (Supplementary Fig. 5j).

In the Class 3 structure we also observe movement of the DNA relative to the entire histone octamer. When compared with the Class LS structure[20], in the Class 3 structure the density for ~ 1 bp of the DNA is missing at one DNA entry/exit site, indicating that the entire DNA moved inward (Fig. 4g and Supplementary Fig. 5k). This is consistent with the direction of the DNA movement at SHL 1 and 2, and shows that in the Class 3 map the DNA moved ~ 1 bp relative to the histone octamer.

The Class 3 structure reveals the mechanism of histone octamer translocation on the DNA. Deformation of the histone octamer in the Class 2 structure generates the strain that leads to translocation of the DNA by the octamer as observed in the Class 3 structure. In the first step of the octamer translocation, H3 α1 translocates to the DNA strand that is normally bound by the H4 α1, whereas H4 α1 detaches from the DNA at SHL 1.5 and translocates to the DNA at SHL 0.5. This pushes the DNA for ~ 1 bp over the histone octamer and shows the first step of DNA sliding by the histone octamer.

**Histone octamer plasticity is required for nucleosome sliding**. Our structures indicate that rearrangements of the histone octamer translocate DNA. To test whether histone octamer plasticity is essential for nucleosome sliding, we have assembled nucleosomes with 227 bp-long DNA that contains strong positioning 601 DNA sequence in the center. Previously, it has been reported that nucleosomes can slide in a non-catalyzed way on various weakly positioning DNA sequences at physiological or slightly elevated temperatures[5,7,8,10]. We have observed that at elevated temperatures, nucleosomes can slide even on the strongly positioning 601 DNA sequence (Fig. 5a, b and Supplementary Fig. 6b, c). To determine whether rearrangements in the histone octamer are required for nucleosome sliding, we have cross-linked the nucleosome. The cross-linked histone octamer is rigid and unable to go through conformational changes. Although the histone octamer was cross-linked and migrates as one band on the denaturing gel (Supplementary Fig. 6a), the octamer was not cross-linked with DNA. The DNA can be disassembled from the cross-linked nucleosomes by increased salt or temperature (Fig. 5c and Supplementary Fig. 6d, e). In thermal shift assays we observed that the native nucleosomes re-position on the long 601 DNA at elevated temperature. The cross-linked histone octamer, however, was not able to move on the DNA and remains on the 601 DNA sequence (Fig. 5a-d and Supplementary Fig. 6b).

Next, we have introduced a site-specific disulfide cross-link between H3 and H4, which was previously shown to impair catalyzed nucleosome sliding[12]. The disulfide bridge between H3 F104C and H4 V43C was sufficient to immobilize histone octamer on DNA and we did not observe non-catalyzed nucleosome sliding of the cross-linked sample (Fig. 5d and Supplementary Fig. 7a, b). The addition of a reducing agent removed the disulfide bridge between H3 and H4, and re-enabled histone octamer sliding on DNA (Fig. 5d).

These data show that plasticity of the histone octamer and rearrangements in the histone octamer are required for non-catalyzed nucleosome sliding.

## Discussion

The NCP has been crystallized numerous time; however, changes in the histone octamer core have not been observed. Our structures show that the rearrangement of the histone octamer is coupled with DNA translocation (Fig. 5e, f). The molecular basis for DNA translocation could be the outcome of two possibilities. The histone octamer deformation could be a consequence of DNA movement and changes in the octamer core subsequently accommodate the DNA in its new conformation. Alternatively, structural changes in the histone octamer might be the driving force for DNA translocation.

Our results show that the histone octamer core is structurally plastic and can adopt multiple distinct conformations. These structural rearrangements of the histone core translocate DNA in an uncatalyzed manner and are required for nucleosome sliding (Fig. 5). Structural rearrangement of the histone core is also used by chromatin remodeling enzymes to mobilize DNA within chromatin[12]. Cross-linking of residues between H3 and H4 has been shown to impair the function of several chromatin remodeling enzymes[12], and our structures show that these residues move apart in the Class 2 and the Class 3 map when compared with the X-ray structure (Supplementary Fig. 7a, b). Cross-linking of these residues would block transition of the histone octamer into the conformation observed in the Class 2 and the Class 3 structures. This indicates that the Class 2 and the Class 3 structures resemble a conformation that is also exploited by ISWI chromatin remodeling enzymes to slide nucleosomes. Recent structure of Snf2-Nucleosome complex shows that Snf2 pulls the DNA out of the nucleosome plane without distorting the histone octamer (Supplementary Fig. 7c)[35]. This is consistent with the finding that cross-linking of residues between H3 and H4 did not affect remodeling by Snf2 enzyme, indicating that these enzymes employ different mechanisms[12]. Most chromatin remodeling enzymes have been shown to bind near SHL 2 where we also observe the most pronounced rearrangements in our structures[36–42]. Our data show conformational changes in the

nucleosome that are also exploited by several chromatin remodeling enzymes. In this simple model, chromatin remodeling enzymes would provide energy to direct nucleosomes to less favorable sequences, analogous to thermal shift.

A recent study has shown that ISWI remodelers translocate ~ 7 bp of DNA out of the exit site before any DNA is translocated inward[43]. This suggested that histone octamer might deform to accommodate less DNA. Our structures show that the histone octamer can deform, which also deforms the DNA, especially at SHL 2 (Fig. 3). Our data show that the histone octamer deformation can compensate for tension generated by missing DNA bps and explains how DNA can be first translocated outward before new DNA is translocated inward[43,44]. In the Class 3 structure, the DNA at SHL 1 and 2 is moved 5 Å relative to histones, indicating a translocation of 1 bp. In agreement, in the Class 3 structure ~ 1 bp of DNA is missing at the DNA entry/exit site from which DNA is translocating.

In our structures we observed that movement of DNA leads to rearrangement of the H4 tail. The H4 tail is essential for enzyme-driven chromatin remodeling, and the chromatin remodelers ISWI and Chd1 require the basic patch of the H4 tail (K16-R17-H18-R19) for the activity[6,40,45–50]. In the Class 1 structure, we observed that the H4 tail interacts with the phosphate backbone of DNA at SHL 2.5. When DNA at SHL 2.5 translocates away (Class 2, 3), the basic patch of the H4 tail cannot reach the DNA at SHL 2.5, and the H4 tail becomes more flexible and inserts into the major groove at SHL 2. This rearrangement of the tail might serve as a signal for the distorted state of the nucleosome for chromatin remodeling complexes and might regulate their binding and activity.

In our recent study ~ 10% of nucleosomes had unwrapped DNA[20]. In the current dataset we observed that ~ 15% of nucleosome particles show DNA unwrapping. The increased proportion of unwrapped nucleosomes is likely because of the increased salt concentration[51]. Although in the previous dataset nucleosomes were at frozen at 50 mM NaCl, in this dataset the sample contained 150 mM salt. At even higher salt concentration, nucleosomes unwrap even more and start to disassemble[20].

In all crystal structures of the NCP, the histone octamer is found in the same conformation[2,3]. In this study we present structures of the nucleosome with a differently organized histone octamer. Our structures show that the nucleosome has considerable structural plasticity at its disposal and can adopt multiple conformations. Structural plasticity of the octamer permits uncatalyzed DNA translocation and is also required for the function of several chromatin remodeling complexes[12]. It is likely

**Table 1 Cryo-EM data collection, refinement, and validation statistics**

| | Class 1<br>EMD-4297<br>PDB ID 6FQ5 | Class 2<br>EMD-4298<br>PDB ID 6FQ6 | Class 3<br>EMD-4299<br>PDB ID 6FQ8 |
|---|---|---|---|
| **Data collection and processing** | | | |
| Magnification | 105 000 | 105 000 | 105 000 |
| Voltage (kV) | 300 | 300 | 300 |
| Electron exposure (e – /Å$^2$) | 100 | 100 | 100 |
| Defocus range (μm) | − 1.0 to − 3.0 | − 1.0 to − 3.0 | − 1.0 to − 3.0 |
| Pixel size (Å) | 1.4 | 1.4 | 1.4 |
| Symmetry imposed | C2 | C2 | C1 |
| Initial particle images (no.) | ~ 410 000 | ~ 410 000 | ~ 410 000 |
| Final particle images (no.) | 51 000 | 58 000 | 39 000 |
| Map resolution (Å) | | | |
| FSC threshold | 3.85 | 4.05 | 4.8 |
| Map resolution | | | |
| range (Å) | 3.7–5.0 | 3.9–5.0 | 4.5–6.0 |
| **Refinement** | | | |
| Initial model used | 3LZ1 | 3LZ1 | 3LZ1 |
| Model resolution (Å) | | | |
| FSC threshold | 3.8 | 4.0 | 4.8 |
| Model resolution | | | |
| range (Å) | 235–3.8 | 235–4.0 | 235–4.8 |
| Map sharpening *B*-factor (Å$^2$) | − 100 | − 100 | − 100 |
| Model composition | | | |
| Non-hydrogen atoms | 12 215 | 11 987 | 11 917 |
| Protein residues | 771 | 752 | 744 |
| Ligands | 0 | 0 | 0 |
| *B* factors (Å$^2$) | | | |
| Protein | 102.30 | 185.18 | 204.17 |
| Ligand | 0 | 0 | 0 |
| R.m.s. deviations | | | |
| Bond lengths (Å) | 0.008 | 0.009 | 0.009 |
| Bond angles (°) | 1.140 | 1.164 | 1.216 |
| Validation | | | |
| MolProbity score | 1.23 | 1.50 | 1.53 |
| Clashscore | 3.63 | 7.97 | 8.49 |
| Poor rotamers (%) | 0.31 | 0.32 | 0.16 |
| Ramachandran plot | | | |
| Favored (%) | 97.63 | 97.69 | 97.66 |
| Allowed (%) | 2.37 | 2.31 | 2.34 |
| Disallowed (%) | 0.0 | 0.0 | 0.0 |

to be that structural rearrangements of the histone octamer have a role beyond DNA translocation and nucleosome remodeling[13,14,17,19,20]. For example, histone variants or some histone modifications might stabilize distinct conformations of the nucleosome and this might be essential for their function. Probably, other chromatin-modifying machineries might also exploit the intrinsic plasticity of the nucleosome for their functions.

## Methods

**Nucleosome reconstitution.** *Xenopus laevis* histones were co-expressed and co-purified as soluble H2A/H2B and (H3/H4)2 histone pairs[23,52,53]. *Escherichia coli* Rosetta cells containing plasmid for histone co-expression were induced with 0.2 mM IPTG overnight at 18 °C. Pelleted cells were resuspended in 50 mM sodium phosphate pH 8.0, 2 M NaCl, 20 mM imidazole, 3 mM β-mercaptoethanol, 1 mM phenylmethylsulfonyl fluoride, and lysed using a French press. The cleared supernatant was incubated with Ni Sepharose 6 Fast Flow resin (GE Healthcare). After binding, the resin was extensively washed and histone proteins eluted with the buffer containing 300 mM imidazole. The histone proteins were further purified on ion-exchange and size-exclussion column[23]. The histone octamer was assembled in 25 mM HEPES/NaOH pH 7.5, 2 M NaCl, 1 mM dithiothreitol (DTT). A 2.8-fold excess of H2A/H2B histone dimer was mixed with H3/H4 histone tetramer and the octamer was purified by size-exclusion chromatography equilibrated in 15 mM HEPES/NaOH pH 7.5, 2 mM NaCl, 1 mM DTT (Supplementary Fig. 1a, b). Histone mutants H3 F104C and H4 V43C were purified in the presence of excess DTT during all the steps of protein purification[12]. To purify the Cysteine variants, only the gel filtration step was carried out. The assembly of Cysteine variants octamer was carried out in the presence of 10 mM freshly made DTT in the buffers.

DNA for nucleosome reconstitution was PCR amplified from a plasmid containing the strong positioning 601 DNA sequence[24]. Oligonucleotides used are listed in Supplementary Table 1. One hundred and forty-nine base pairs (149 bp) of 601 DNA was used for nucleosome reconstitution for cryo-EM structures. For sliding assay, 227 bp 601 DNA was used (± 40 bp on each side or + 80 bp on one side of the 601 sequence). PCR products were purified by phenol chloroform extraction. After ethanol precipitation, DNA was resuspended in 15 mM HEPES/NaOH pH 7.5, 2 M NaCl, 1 mM DTT. The nucleosome reconstitution was done by 'double bag' dialysis[23,54]. The dialysis buttons, containing 0.25 ml of the histone octamer:DNA mixture in 2 M salt buffer, were placed inside a dialysis bag, filled with 50 ml of the size-exclusion buffer. The dialysis bag was immersed into a 1 liter of buffer containing 15 mM HEPES/NaOH pH 7.5, 1 M NaCl, 1 mM DTT, and dialysed overnight at + 4 °C. The next day, the dialysis bag, containing 50 ml of 1 M salt buffer and the dialysis buttons, was immersed into a 1 liter low-salt buffer (100 mM NaCl, 15 mM HEPES/NaOH pH 7.5, 1 mM DTT) and dialysed for 5–6 h. At the end of the buffer exchange process, the final buffer concentration was 150 mM NaCl, 15 mM HEPES/NaOH pH 7.5, 1 mM DTT.

**Nucleosome cross-linking.** The glutaraldehyde was added to a final concentration of 0.1% (v/v) and the samples were incubated for 5 min at the room temperature for cryo-EM grid preparation. The cross-linking was quenched with 50 mM Tris/HCl pH 7.5 (final concentration) and incubated at least 30 min at + 4 °C. The samples were concentrated to 2 mg ml⁻¹ for cryoEM grids preparation.

For nucleosome sliding assays, samples were cross-linked for 30 min at + 4 °C, to reduce any histone octamer movements prior the assay. The cross-linking was quenched with 50 mM Tris/HCl pH 7.5 (final concentration).

Disulfide cross-linking was done by extensive dialysis of the NCP under reducing conditions. The NCP containing H3 F104C and H4 V43C mutations was assembled by 'double bag' dialysis under oxidizing conditions (final dialysis buffer: 15 mM HEPES pH 7.5, 1 mM DTT). After the assembly, three dialysis steps under reducing conditions (the dialysis buffer: 15 mM HEPES pH 7.5) were carried out. Two dialysis steps were done overnight.

**Nucleosome sliding.** For thermal shift assays, 10 µl of each nucleosome sample were incubated at 60 °C (227 bp-long 601 DNA sequence). The time points were taken as indicated. The site-specific disulfide cross-link between H3 F104C and H4 V43C was removed by adding 5 mM DTT (final concentration) and incubating the sample 1 h on ice before the thermal shift assay. The thermal shift assays were analyzed on a 5% native PAGE. The gel was run in 0.2 × TBE buffer at 200 V for 75 min in the cold room. The gel was stained with SYBR Gold.

**Thermal disassembly of the nucleosome.** Ten microliters of each nucleosome sample (40 ng µl⁻¹, 15 mM HEPES pH 7.5, 1 mM DTT) was incubated on ice or at the indicated temperatures (70, 74, 78, 81, 84, 86, or 90 °C) in a PCR machine for 15 min. The glycerol (final concentration 4% v/v) was added to the samples and they were analysed using 6% native PAGE. The gel was run in 1 × TBE buffer at 150 V for 70 min in the cold room. The gel was stained with SYBR Gold.

**Salt disassembly of the nucleosome.** Ten microliters of each nucleosome sample (40 ng µl⁻¹, 15 mM HEPES pH 7.5, 1 mM DTT) was supplemented with the buffer containing additional NaCl so that the final salt concentration is as indicated on the Figures (0, 0.1, 0.2, 0.3, 0.5, 0.75, 1, or 1.5 M NaCl). The samples were incubated at 25 °C for 30 min. The glycerol (final concentration 4% v/v) was added to the samples and they were analysed using 6% native PAGE. The gel was run in 1 × TBE buffer at 150 V for 70 min in the cold room. The gel was stained with SYBR Gold.

**Cryo-EM grid preparation and data collection.** We used Quantifoil R2/1 holey carbon grids. The sample was vitrified with Leica EM GP automatic plunge freezer. Temperature was kept at + 15 °C and the humidity at 95%. Three microliters of NCP sample were applied to freshly glow-discharged grid, blotted for 3 s, and plunge-frozen in the liquid ethane. The data were recorded on a FEI Titan Halo (FEI) at 300 kV with a Falcon 2 direct electron detector (FEI) (750 micrographs) (MPI for Biochemistry, Martinsried, Germany). The nominal magnification was 75 000 resulting in an image pixel size of 1.4 Å per pixel on the object scale. Data were collected in a defocus range of 10 000–30 000 Å with a total exposure of 100 e Å⁻². Forty frames were collected and aligned using the Unblur software package using a dose filter[55].

Several thousand particles were manually picked and carefully cleaned in XMIPP[56] to remove inconsistent particles. The resulting useful particles were used for semi-automatic and automatic particle picking in XMIPP. The contrast transfer function parameters were determined using CTFFIND4[57]. The 2D class averages were generated with Relion software package[58] and inconsistent class averages were removed from further data analysis. The three-dimensional refinements and classifications were done in Relion. Particles were split into two datasets and refined independently, and the resolution was determined using the 0.143 cutoff (Relion auto-refine option). Local resolution was determined with Relion 2.0. and all maps were filtered to the local resolution with a B-factor determined by Relion. The initial reference was filtered to 60 Å in Relion. C2 symmetry was applied during refinements for Class 1 and Class 2, whereas Class 3 was refined with C1 symmetry.

In our data sets, most particles were in the top view as observed by angular distribution. To exclude the possibility that orientation bias might lead to the distortion of the Class 2 structure, we have manually selected classes to enrich for disk views. In the Class 2A, 50% of particles are in the disk orientation (Supplementary Fig. 7d, e). This particles were reconstructed to 6.5 Å and the resulting map resembles Class 2 structure (Supplementary Fig. 7d-f). The Class 2 model fits well into this map, whereas the X-ray model does not fit (Supplementary Fig. 7f). This shows that distortion of the nucleosome observed in the Class 2 structure is not a result of orientation bias.

Molecular models were built using Coot[59] and refined in Phenix[60]. Visualization of all cryo-EM maps was done with Chimera[61]. The Chimera software package was used for superposition of Class 1–3 maps and rigid body fitting of models into superimposed maps. The movements were determined by calculating RMSD of Cα. We have build independent models for two half data sets, to determine the accuracy and uncertainty of models atoms from the models built with the combined maps were randomly displaced to 0.5 Å, and refined against one of two half maps obtained from independent half datasets. To show uncertainty, model variation was calculated in Chimera as RMSD of Cα backbone between two models (half1 and half2 model) (Supplementary Fig. 3f and 5e). These data show that for Class 1 and Class 2 uncertainty is < 0.5 Å for most of the model. For Class 3 uncertainty is < 1.0 Å for most regions.

**Data availability.** EM densities have been deposited in the Electron Microscopy Data Bank under accession codes EMD-4297 (Class 1), EMD-4298 (Class 2), and EMD-4299 (Class 3). The coordinates of EM-based models have been deposited in the Protein Data Bank under accession codes PDB 6FQ5 (Class 1), 6FQ6 (Class 2), and 6FQ8 (Class 3). All other data are available from the corresponding author upon reasonable request.

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

## Acknowledgements

We thank Elena Conti and the Cryo-EM facility at Max Planck Institute for Biochemistry in Martinsried for access to cryo-EM microscopes. Without their support, this work would not be possible. We thank Sigrun Jaklin for excellent technical assistance, Segolene Demolin for the help with the histone purifications, and Nikolai Czech for the help with the DNA purifications. Plasmids containing the cysteine variant histone proteins were generous gift of Dr. G. Narlikar. We also thank the Halic lab for comments on the manuscript. This work was supported by the ERC-smallRNAhet-309584.

## Author contributions

S.B. and M.H. designed the experiments. S.B. performed biochemical experiments and electron microscopy. M.S. assisted with electron microscopy. S.B. and M.H. analyzed the data. S.B. and M.H. wrote the paper.

## Additional information

**Competing interests:** The authors declare no competing interests.

