## [Peer Review File(PDF 276 kb) · Nature Communications]

Reviewers' comments:

Reviewer #1 (Remarks to the Author):

In this study, Bilokapic et al have reported the structures of nucleosomes with altered conformations under physiological salt concentrations using cryo-EM. In low salt, they observe nucleosomes that are structurally similar to the crystal structure. In physiological salt conditions, they detect three structural classes of nucleosome, of which class 1 is similar to the crystal structure, but classes 2 and 3 show noticeable structural differences. The differences resulting from rearrangement of different histone helices and concomitant DNA movements lead to contraction along the dyad axis and expansion along the perpendicular axis. Together, the structural changes can help explain DNA mobility on nucleosomes as both class 2 and 3 structures show rearrangements of DNA near the SHL2 region. Further, the authors show that glutaraldehyde cross-linking of histone octamer inhibits thermal sliding of DNA on nucleosomes, suggesting that the mobility of histone segments plays a key role in thermal DNA sliding.

Previous work from the Kadonaga group has identified conformational isomers of nucleosomes that protect 80 vs. 147 bp of DNA (prenucleosome) and work from the Narlikar group has shown that ISWI enzymes can distort the histone core. The Bilokapic et. al. work substantially adds to the current discussion of nucleosome plasticity by providing the first high resolution structures of conformationally altered nucleosomes. These structures represent an important advance for understanding how alterations in nucleosome conformation can be used to regulate genome accessibility.

The work is suitable for publication in Nature Communications after the following issues are addressed.

1. For the gel in Fig 5a, some redistribution of the cross-linked nucleosomes is seen as a faster migrating smear. It is possible that this represents sliding and that the migration of the slid nucleosomes is affected due to the cross-linking. To ensure that there is no sliding comparable to WT nucleosomes, the authors should generate end-positioned cross-linked nucleosomes as standards to run next to the sliding reactions.

2. It is important that the authors distinguish between saying they have discovered the phenomenon of nucleosome plasticity (which has been discovered previously) vs. saying they have obtained the first structures of distorted nucleosomes (which is exciting in its own right). The specific places where this should be clarified and some suggestions for edits are listed below.

(i) on page 3, the authors say “ It has also been proposed that chromatin remodelers may be able to conformationally rearrange the histone octamer when they translocate DNA”.

This has not only been proposed but also demonstrated by the NMR studies carried out in Sinha et al, 2017. Other prior studies have also provided evidence for nucleosome plasticity (C. P. Prior, C. R. Cantor, E. M. Johnson, V. C. Littau, V. G. Allfrey, Cell 34, 1033–1042, 1983 & J. Fei et al. Genes Dev. 29, 2563–2575, 2015)

Hence the sentence could be modified and extended to say “It has been shown that ISWI chromatin remodelers can conformationally rearrange the histone octamer and such rearrangement is essential for

translocating DNA. These results are consistent with prior observations showing conformational flexibility of a nucleosome in the context of transcription and nucleosome assembly.”

(ii) Page 3 “We found that nucleosomes are highly plastic and that conformational changes in the histone octamer translocate DNA and promote nucleosome sliding. This intrinsic property of the nucleosome is also utilized by chromatin remodeling enzymes”

This sentence can be modified to:

“Our structures uncover the types of conformational changes possible in a nucleosome. These conformational changes can explain DNA translocation and are essential to promote thermally driven nucleosome sliding. We propose that the same intrinsic property of the nucleosome is also utilized by chromatin remodeling enzymes”

(iii) Page 5, “Our data show that the nucleosome can adopt multiple conformations and is more plastic than currently considered.”

This sentence can be modified to:

“Our data show that the nucleosome can adopt multiple conformations providing a structural basis for nucleosome plasticity.”

(iv) Page 10, “Our results show that the histone octamer core is structurally plastic and can accommodate multiple conformations.”

This sentence can be modified to:

“Our results show that the histone octamer core can adopt multiple distinct conformations.”

(v) Page 11, “It is likely that structural rearrangements of the histone octamer play a role beyond DNA translocation and nucleosome remodeling.”

Here reference also the Allfrey and Kadonaga papers in (i) above.

3. It will help the reader if the authors can show an overlap of the class 1 and 2 structures to make the contraction/expansion along the two axes more obvious.

4. While the structural changes seen in the work are suggestive of a possible lever like role of histone helices, there is no biochemical or kinetic data that show the proposed lever activity of the histone helices. Therefore, in the ‘structural rearrangement of the histone octamer’ section, the sentence: ‘our data show that H3 $\alpha 2$ serves as a lever...’ - should be rephrased to: “Our data raise the possibility that H3 $\alpha 2$ serves as a lever that pushes the DNA at one end and pulls the DNA at the opposite end (Fig. 2b, c and Supplementary Fig. 4b)”

5. In the class 3 structure, the authors observe differences between the two halves of nucleosomes, which they classify as side A and B. Is there any explanation for this? For example, is the asymmetry of

the 601-nucleosome structure playing any role? Also the A and B sides were not easy to locate. Can these be labeled on the figures?

6. In supplementary fig. 2g, the local resolution is lower for the class 2 structure compared to class 1 structure. Is it an outcome of presence of increased motional dynamics in nucleosome structure in the class 2 form? The authors should comment on it.

7. Is the y-axis in Fig. 5b simply a ratio of the two states? A better measure would be to plot the fraction slid ($[\text{end positioned}]/([\text{end positioned}]+[\text{centered}])$).

8. In class 3, the authors see one base pair of DNA moved into the core at the entry site. It appears from the writing that they are suggesting this extra base pair is accommodated within the core. Can they rule out an extra base-pair coming out from the exit site?

9. Define the red, grey and purple colours in Figure 5d&e.

10. Why do the 0.1M NaCl lanes look different between Supplementary Figure 6c and Figure 5c? Specifically, why does the DNA alone band run much faster in Supplementary Figure 6c?

11. What is the *DNA band in Supplementary Figure 6d?

Reviewer #2 (Remarks to the Author):

Bilokapic et al., have investigated structures of nucleosomes determined by cryo-EM that exist in different conformations in vitrified ice. Furthermore, the paper presents a detailed comparison of new and previously determined cryo-EM and X-ray structures of the nucleosome. The comparison is interesting for structural biologist working in the field of histone-DNA interactions. To my surprise a very similar article, by the same group, was published in NSMB only a few days ago. Although the abstract reads slightly differently, the structures and analysis is similar.

The submitted manuscript is therefore more like an extended view suitable for a specialised journal or possibly a current opinion paper. Therefore, I can not recommend the manuscript for publication in nature communications.

Reviewer #1 (Remarks to the Author):

In this study, Bilokapic et al have reported the structures of nucleosomes with altered conformations under physiological salt concentrations using cryo-EM. In low salt, they observe nucleosomes that are structurally similar to the crystal structure. In physiological salt conditions, they detect three structural classes of nucleosome, of which class 1 is similar to the crystal structure, but classes 2 and 3 show noticeable structural differences. The differences resulting from rearrangement of different histone helices and concomitant DNA movements lead to contraction along the dyad axis and expansion along the perpendicular axis. Together, the structural changes can help explain DNA mobility on nucleosomes as both class 2 and 3 structures show rearrangements of DNA near the SHL2 region. Further, the authors show that glutaraldehyde cross-linking of histone octamer inhibits thermal sliding of DNA on nucleosomes, suggesting that the mobility of histone segments plays a key role in thermal DNA sliding.

Previous work from the Kadonaga group has identified conformational isomers of nucleosomes that protect 80 vs. 147 bp of DNA (prenucleosome) and work from the Narlikar group has shown that ISWI enzymes can distort the histone core. The Bilokapic et. al. work substantially adds to the current discussion of nucleosome plasticity by providing the first high resolution structures of conformationally altered nucleosomes. These structures represent an important advance for understanding how alterations in nucleosome conformation can be used to regulate genome accessibility.

The work is suitable for publication in Nature Communications after the following issues are addressed.

This study and the study published recently in NSMB are completely different, although, the data might look similar at the first look (similar images, similar colors, similar style) since the figures were done by the same authors. The structures are different and also biological processes that we address are different.

In the study published in NSMB we show 9 cryo EM structures that show how DNA unwraps from the nucleosome, which leads to H2A/H2B dissociation and hexasome formation. We also show that H2A/H2B flexibility is required for nucleosome stability. In this study we show 3 structures of fully wrapped nucleosome (canonical, distorted and translocated) that give first insight into non-catalyzed movement of histone octamer on the DNA. None of these structures are published in the NSMB paper. These structures come from a different dataset of nucleosomes frozen under more physiological salt conditions (150 mM NaCl). Nucleosomes in NSMB paper were frozen at 50 mM NaCl. The Class LS structure which is shown in the supplemental information actually originates from the published dataset (NSMB) and we will reference it properly. Classes 1-3 are new structures that come from an independent dataset, frozen at different conditions.

Our structure of the distorted nucleosome (Class 2) shows that fully wrapped nucleosome can adopt multiple states in solution at physiological conditions. These states might be stabilized by histone modifications, histone variants or extrinsic factors. Structure of the translocated nucleosome (Class 3) shows for the first time how histone octamer moves on the DNA and provides mechanistic insights into DNA translocation. In the Class 3 structure several alpha-helices interact with different DNA strand, indicating their translocation. Comparison of Class 1-3 structures shows conformational changes in the histone octamer that explain how histone octamer translocates on DNA. The conformational changes

we describe here are different than conformational changes described during the process of DNA unwrapping (NSMB). Also the complementary biochemical assays are different in these two studies. In addition, we have included our new data with nucleosomes containing site specific disulfide cross-link between H3 and H4 as described in Sinha et al (see new Figure 5d). These data show that cross-linking H3 and H4 is sufficient to immobilize nucleosome on the DNA and is consistent with our structural observations.

These two studies show different structures and describe two different biological processes: nucleosome sliding and DNA unwrapping. We do not think that these studies overlap.

We have another manuscript in review that describes nucleosome structures. This manuscript reports interactions between two nucleosomes and is not related.

1. For the gel in Fig 5a, some redistribution of the cross-linked nucleosomes is seen as a faster migrating smear. It is possible that this represents sliding and that the migration of the slid nucleosomes is affected due to the cross-linking. To ensure that there is no sliding comparable to WT nucleosomes, the authors should generate end-positioned cross-linked nucleosomes as standards to run next to the sliding reactions.

We have generated end positioned native and cross-linked nucleosomes. Both native and cross-linked end positioned nucleosomes migrate the same. This shows that cross-linking does not affect migration of nucleosomes (Supplementary Figure 6c).

We have also used the histone proteins with introduced cysteins in H3 and H4 previously described by Sinha et al. We have introduced the disulfide bridge between H3F104C and H4V43C which was sufficient to immobilize nucleosome. When we have removed the disulfide bridge under reducing conditions (DTT), the nucleosome became mobile again. These data nicely show that cross-linking H3 and H4 is sufficient to immobilize histone octamer on the DNA (Figure 5d).

2. It is important that the authors distinguish between saying they have discovered the phenomenon of nucleosome plasticity (which has been discovered previously) vs. saying they have obtained the first structures of distorted nucleosomes (which is exciting in its own right). The specific places where this should be clarified and some suggestions for edits are listed below.

(i) on page 3, the authors say “ It has also been proposed that chromatin remodelers may be able to conformationally rearrange the histone octamer when they translocate DNA”.

This has not only been proposed but also demonstrated by the NMR studies carried out in Sinha et al, 2017. Other prior studies have also provided evidence for nucleosome plasticity (C. P. Prior, C. R. Cantor, E. M. Johnson, V. C. Littau, V. G. Allfrey, Cell 34, 1033–1042, 1983 & J. Fei et al. Genes Dev. 29, 2563–2575, 2015)

Hence the sentence could be modified and extended to say “It has been shown that ISWI chromatin remodelers can conformationally rearrange the histone octamer and such rearrangement is essential for translocating DNA. These results are consistent with prior observations showing conformational flexibility of a nucleosome in the context of transcription and nucleosome assembly.”

We have modified this sentence accordingly and also referenced previous and our recent work showing

nucleosome flexibility during DNA unwrapping.

This and other modifications are labeled red in the new version of the manuscript.

(ii) Page 3 “We found that nucleosomes are highly plastic and that conformational changes in the histone octamer translocate DNA and promote nucleosome sliding. This intrinsic property of the nucleosome is also utilized by chromatin remodeling enzymes”

This sentence can be modified to:

“Our structures uncover the types of conformational changes possible in a nucleosome. These conformational changes can explain DNA translocation and are essential to promote thermally driven nucleosome sliding. We propose that the same intrinsic property of the nucleosome is also utilized by chromatin remodeling enzymes”

We have modified this sentences.

(iii) Page 5, “Our data show that the nucleosome can adopt multiple conformations and is more plastic than currently considered.”

This sentence can be modified to:

“Our data show that the nucleosome can adopt multiple conformations providing a structural basis for nucleosome plasticity.”

We have modified this sentence.

(iv) Page 10, “Our results show that the histone octamer core is structurally plastic and can accommodate multiple conformations.”

This sentence can be modified to:

“Our results show that the histone octamer core can adopt multiple distinct conformations.”

We have modified this sentence.

(v) Page 11, “It is likely that structural rearrangements of the histone octamer play a role beyond DNA translocation and nucleosome remodeling.”

Here reference also the Allfrey and Kadonaga papers in (i) above.

We have referenced previous work.

3. It will help the reader if the authors can show an overlap of the class 1 and 2 structures to make the contraction/expansion along the two axes more obvious.

We have tried this, but the figure doesn't look very informative as nucleosome structure is too compact and it is hard to see changes because of many overlaps.

4. While the structural changes seen in the work are suggestive of a possible lever like role of histone helices, there is no biochemical or kinetic data that show the proposed lever activity of the histone helices. Therefore, in the 'structural rearrangement of the histone octamer' section, the sentence: 'our data show that H3 α 2 serves as a lever...' - should be rephrased to: "Our data raise the possibility that H3 α 2 serves as a lever that pushes the DNA at one end and pulls the DNA at the opposite end (Fig. 2b, c and Supplementary Fig. 4b)"

We have modified this accordingly.

5. In the class 3 structure, the authors observe differences between the two halves of nucleosomes, which they classify as side A and B. Is there any explanation for this? For example, is the asymmetry of the 601-nucleosome structure playing any role? Also the A and B sides were not easy to locate. Can these be labeled on the figures?

601 nucleosome is not symmetrical, but at this resolution we are not able to determine which DNA sequence is present on which side of the nucleosome. It is highly likely that DNA sequence plays a role. We observe that histone alpha-helices flipped back first on one side of the nucleosome, leading to asymmetry in histone octamer conformation. We believe that this is due to DNA sequence.

We have labeled side A and side B better in figures. We have also included new panel showing side A and B on the nucleosome (Supplementary Fig. 5f)

6. In supplementary fig. 2g, the local resolution is lower for the class 2 structure compared to class 1 structure. Is it an outcome of presence of increased motional dynamics in nucleosome structure in the class 2 form? The authors should comment on it.

We think that class 2 is slightly more dynamic leading to a lower overall resolution. Although the difference in the resolution is very small suggesting that Class 2 structure is also a pretty stable conformation. We also found particles that reconstruct to structures that appear to be somewhere in between Class 1 and Class 2 maps. However, these structures are at lower resolution suggesting that these states are more transient and less defined.

7. Is the y-axis in Fig. 5b simply a ratio of the two states? A better measure would be to plot the fraction slid ($[\text{end positioned}]/([\text{end positioned}] + [\text{centered}])$).

We have changed this to the suggested plot.

8. In class 3, the authors see one base pair of DNA moved into the core at the entry site. It appears from the writing that they are suggesting this extra base pair is accommodated within the core. Can they rule out an extra base-pair coming out from the exit site?

We cannot rule this out. We do not see the extra density for additional base-pair, but it could be that this base pair is more flexible. Actually, we try not to suggest anything and would like to leave both possibilities open.

9. Define the red, grey and purple colours in Figure 5d&e.

We have defined this.

10. Why do the 0.1M NaCl lanes look different between Supplementary Figure 6c and Figure 5c? Specifically, why does the DNA alone band run much faster in Supplementary Figure 6c?

The gel in the Supplementary Figure 6c (in the new version 6d) was simply run for a longer time, leading to larger separation of DNA and nucleosomes.

11. What is the *DNA band in Supplementary Figure 6d?

It is a DNA band we observe at elevated temperatures. This is probably a partially melted DNA.

Reviewer #2 (Remarks to the Author):

Bilokapic et al., have investigated structures of nucleosomes determined by cryo-EM that exist in different conformations in vitrified ice. Furthermore, the paper presents a detailed comparison of new and previously determined cryo-EM and X-ray structures of the nucleosome. The comparison is interesting for structural biologist working in the field of histone-DNA interactions. To my surprise a very similar article, by the same group, was published in NSMB only a few days ago. Although the abstract reads slightly differently, the structures and analysis is similar.

The submitted manuscript is therefore more like an extended view suitable for a specialised journal or possibly a current opinion paper. Therefore, I can not recommend the manuscript for publication in nature communications.

The data might look similar at the first look (similar images, similar colors, similar style) since the figures were done by the same authors, but these studies are completely different. The structures are different and also biological processes that we address are different.

In the study published in NSMB we show 9 cryo EM structures that show how DNA unwraps from the nucleosome, which leads to H2A/H2B dissociation and hexasome formation. We also show that H2A/H2B flexibility is required for nucleosome stability. In this study we show 3 structures of fully wrapped nucleosome (canonical, distorted and translocated) that give first insight into non-catalyzed movement of histone octamer on the DNA. None of these structures are published in the NSMB paper. These structures come from a different dataset of nucleosomes frozen under more physiological salt conditions (150 mM NaCl). Nucleosomes in NSMB paper were frozen at 50 mM NaCl. The Class LS structure which is shown in the supplemental information actually originates from the published dataset (NSMB) and we will reference it properly. Classes 1-3 are new structures that come from an independent dataset, frozen at different conditions.

Our structure of the distorted nucleosome (Class 2) shows that fully wrapped nucleosome can adopt multiple states in solution at physiological conditions. These states might be stabilized by histone modifications, histone variants or extrinsic factors. Structure of the translocated nucleosome (Class 3) shows for the first time how histone octamer moves on the DNA and provides mechanistic insights into DNA translocation. In the Class 3 structure several alpha-helices interact with different DNA strand, indicating their translocation. Comparison of Class 1-3 structures shows conformational changes in the histone octamer that explain how histone octamer translocates on DNA. The conformational changes we describe here are different than conformational changes described during the process of DNA

unwrapping (NSMB). Also the complementary biochemical assays are different in these two studies. In addition, we have included our new data with nucleosomes containing site specific disulfide cross-link between H3 and H4 as described in Sinha et al (see new Figure 5d). These data show that cross-linking H3 and H4 is sufficient to immobilize nucleosome on the DNA and is consistent with our structural observations.

These two studies show different structures and describe two different biological processes (nucleosome sliding and DNA unwrapping). We do not think that these studies overlap.

Reviewers' comments:

Reviewer #1 (Remarks to the Author):

The necessary cross-linking controls and additional S-S cross-linking experiments add to the significance of the structures in this work. I also now mostly understand the differences between their NSMB paper and the data presented here. I have one remaining set of questions about the NSMB paper comparison:

1. In the NSMB work they also looked at structures at 250 mM NaCl. Though these are at a lower resolution, they see DNA unpeeling here, as well as loss of H2A/H2B (Fig. 5). Did they not see octamer distortion in these structures?
2. Correspondingly in the 150 mM NaCl data set (i.e., the work submitted to Nature Communications) I would expect the authors to see DNA unwrapping. Do they see this? If so, they should comment on it so the reader is not confused (see point below).
3. Overall given that the current manuscript has data collected at 150 mM NaCl and the published NSMB work has data collected at 50 and 250 mM NaCl, the authors should very clearly in the discussion compare the results from these three salt concentrations and comment on the differences. With a salt dependence series I would expect a progressive change in the distribution of nucleosome states. In other words I would expect to see the amount of unwrapped nucleosomes to increase at 150 mM NaCl compared to 50 mM NaCl in addition to seeing the wrapped and distorted nucleosomes that the authors see. Instead, if one is to read the NSMB paper and this manuscript one gets the impression that there is a change from unwrapped nucleosomes at 50 mM NaCl to non-unwrapped but distorted nucleosomes at 150 mM NaCl to more unwrapped and disassembling nucleosomes at 250 mM NaCl.

Minor comment:

In my original minor comment # I said:

(iv) Page 10, "Our results show that the histone octamer core is structurally plastic and can accommodate multiple conformations."

This sentence can be modified to:

"Our results show that the histone octamer core can adopt multiple distinct conformations."

[Authors response: We have modified this sentence.]

The sentence is not modified in the revised manuscript.

Reviewers' comments:

Reviewer #1 (Remarks to the Author):

The necessary cross-linking controls and additional S-S cross-linking experiments add to the significance of the structures in this work. I also now mostly understand the differences between their NSMB paper and the data presented here. I have on remaining set of questions about the NSMB paper comparison:

1. In the NSMB work they also looked at structures at 250 mM NaCl. Though these are at a lower resolution, they see DNA unpeeling here, as well as loss of H2A/H2B (Fig. 5). Did they not see octamer distortion in these structures?

The resolution of these maps is too low to see octamer distortion. Only in one map from this dataset we can partially resolve alpha helices, but DNA is poorly resolved and major and minor grooves are not visible. The resolution of maps is too low to make any conclusions regarding octamer distortion.

2. Correspondingly in the 150 mM NaCl data set (i.e., the work submitted to Nature Communications) I would expected the authors to see DNA unwrapping. Do they see this? If so, they should comment on it so the reader is not confused (see point below).

We do see DNA unwrapping in this dataset. We did not analyze this very carefully since structures are quite similar to maps published in NSMB paper. Approximately ~15% of particles show different states of DNA unwrapping.

We have now included this statement in the manuscript.

3. Overall given that the current manuscript has data collected at 150 mM NaCl and the published NSMB work has data collected at 50 and 250 mM NaCl, the authors should very clearly in the discussion compare the results from these three salt concentrations and comment on the differences. With a salt dependence series I would expect a progressive change in the distribution of nucleosome states. In other words I would expect to see the amount of unwrapped nucleosomes to increase at 150 mM NaCl compared to 50 mM NaCl in addition to seeing the wrapped and distorted nucleosomes that the authors see. Instead, if one is to read the NSMB paper and this manuscript one gets the impression that there is a change from unwrapped nucleosomes at 50 mM NaCl to non-unwrapped but distorted nucleosomes at 150 mM NaCl to more unwrapped and disassembling nucleosomes at 250 mM NaCl.

We do see DNA unwrapping at all salt concentrations. At 150 mM salt, we see similar states of nucleosome unwrapping like at 50 mM salt, but we observe a bit higher fraction of unwrapped particles (~15% compared to ~10%). At 250 mM salt we start seeing further DNA unwrapping and nucleosome disassembly. We have included discussion in the manuscript.

Minor comment:

In my original minor comment # I said:

(iv) Page 10, "Our results show that the histone octamer core is structurally plastic and can accommodate multiple conformations."

This sentence can be modified to:

“Our results show that the histone octamer core can adopt multiple distinct conformations.”

[Authors response: We have modified this sentence.]

The sentence is not modified in the revised manuscript.

The sentence was modified but not exactly as suggested by the reviewer. We find that the sentence is fine as it is.